# Pathological, Morphological, Cytogenomic, Biochemical and Molecular Data Support the Distinction between *Colletotrichum cigarro*
*comb. et stat. nov.* and *Colletotrichum kahawae*

**DOI:** 10.3390/plants9040502

**Published:** 2020-04-14

**Authors:** Ana Cabral, Helena G. Azinheira, Pedro Talhinhas, Dora Batista, Ana Paula Ramos, Maria do Céu Silva, Helena Oliveira, Vítor Várzea

**Affiliations:** 1Linking Landscape, Environment, Agriculture and Food, Instituto Superior de Agronomia, Universidade de Lisboa, 1349-017 Lisbon, Portugal; anacabral@isa.ulisboa.pt (A.C.); hmga@edu.ulisboa.pt (H.G.A.); dccastro@fc.ul.pt (D.B.); pramos@isa.ulisboa.pt (A.P.R.); mariaceusilva@isa.ulisboa.pt (M.d.C.S.); heloliveira@isa.ulisboa.pt (H.O.); vitorvarzea@isa.ulisboa.pt (V.V.); 2Centro de Investigação das Ferrugens do Cafeeiro, Instituto Superior de Agronomia, Universidade de Lisboa, 2780-505 Oeiras, Portugal; 3Centre for Ecology, Evolution and Environmental Changes, Faculdade de Ciências, Universidade de Lisboa, 1749-016 Lisbon, Portugal; 4Laboratório de Patologia Vegetal “Veríssimo de Almeida”, Instituto Superior de Agronomia, Universidade de Lisboa, 1349-017 Lisbon, Portugal

**Keywords:** taxonomy, speciation, *Colletotrichum cigarro*, *Colletotrichum kahawae*, Coffee Berry Disease

## Abstract

The genus *Colletotrichum* has witnessed tremendous variations over the years in the number of species recognized, ranging from 11 to several hundreds. Host-specific fungal species, once the rule, are now the exception, with polyphagous behavior regarded as normal in this genus. The species *Colletotrichum kahawae* was created to accommodate the pathogens that have the unique ability to infect green developing coffee berries causing the devastating Coffee Berry Disease in Africa, but its close phylogenetic relationship to a polyphagous group of fungi in the *C. gloeosporioides* species complex led some researchers to regard these pathogens as members of a wider species. In this work we combine pathological, morphological, cytogenomic, biochemical, and molecular data of a comprehensive set of phylogenetically-related isolates to show that the Coffee Berry Disease pathogen forms a separate species, *C. kahawae*, and also to assign the closely related fungi, previously in *C. kahawae* subsp. *cigarro*, to a new species, *C. cigarro comb. et stat. nov*. This taxonomic clarification provides an opportunity to link phylogeny and functional biology, and additionally enables a much-needed tool for plant pathology and agronomy, associating exclusively *C. kahawae* to the Coffee Berry Disease pathogen.

## 1. Introduction

The genus *Colletotrichum* Corda comprises diverse plant pathogens, causing important diseases collectively known as anthracnose. The taxonomy of species in this genus has changed several times during the last decades. Since it was first described, and for over one century, a growing number of species was assigned to *Colletotrichum*. Von Arx [1] listed around 750 species, many of which had been described based on the host plant range rather than on morphology, and drastically reduced this number to 11 species based on morphological characteristics. Sutton [2] revised the genus and recognized 22 species, most of them polyphagous, based on morpho-cultural criteria. In this process, the species *Colletotrichum gloeosporioides* has played a central role in the genus *Colletotrichum*, with numerous pathogens causing important diseases assigned to it. In fact, *C. gloeosporioides* was for a long time considered a species complex, frequently regarded as a dumping taxon for diverse *Colletotrichum* fungi and, in this sense, of little biological, taxonomical, or pathological meaning [3]. More recently, the taxonomy of *Colletotrichum* has witnessed major changes, comprising nearly 200 accepted species [4,5,6], the vast majority of which clustered into species complexes. The *C. gloeosporioides* species complex [7] is the widest, comprising at least 38 species [4].

Although clustering within the *C. gloeosporioides* species complex [8], the species *C. kahawae* Waller and Bridge was recognized as a separate taxon due to the unique feature of its members as being capable of infecting green coffee berries, thus causing the Coffee Berry Disease (CBD) [9]. This disease is restricted to Africa, where it represents the main constraint to sustainable production of Arabica coffee, due to yield losses up to 80% without chemical control [10,11,12,13,14]. The risk of its introduction on the main Arabica growing countries (in America and Asia) makes this a quarantine pathogen, referred to as a biological weapon in Australia [13,14,15].

Other *Colletotrichum* spp., namely other species from the *C. gloeosporioides* complex and *C. acutatum*, are frequently isolated from ripe coffee berries, but they are incapable of causing CBD [9,16,17]. In fact, *C. kahawae* was created to resolve the ambiguity caused by the inclusion of both CBD-causing and non-CBD causing isolates in the previously described species *C. coffeanum* Noak [9,16,18,19]. Additionally, *C. kahawae* isolates are unable to utilize citrate or tartrate as sole carbon sources, while the non-CBD causing isolates from coffee metabolized one or both [9]. However, a group of *Colletotrichum* isolates from multiple hosts and diverse geographic origins (unrelated to coffee) was shown to be indistinguishable from the CBD pathogens for six nuclear gene regions that are usually employed for taxonomic purposes, actin (*act*), calmodulin (*cal*), chitin synthase (*chs1*), glyceraldehyde-3-phosphate dehydrogenase (*gapdh*), manganese-superoxide dismutase (*sod2*), and β-tubulin 2 (*tub2*), along with the rDNA-ITS region [7], in spite of a clear differentiation provided by the mating type gene mat1-2-1 (*mat1-2-1*), a fragment of DNA lyase Apn2 [20] (*apn25L*), and glutamine synthetase (*gs*) [7], as well as the capacity to utilize either citrate or tartrate as a sole carbon source [7,9,21]. In this context, both CBD-causing and the phylogenetically close non-CBD causing isolates were all placed under *C. kahawae* [7], being created *C. kahawae* subsp. *kahawae* to accommodate the CBD-causing isolates and *C. kahawae* subsp. *cigarro* (as *C. kahawae* subsp. “*ciggaro*” B. Weir and P.R. Johnst.) for the phylogenetically close non-CBD causing isolates.

Subsequently, Doyle et al. [22] proposed two new species, *C. fructivorum* and *C. temperatum*, and epitypified the species *C. rhexiae*, all of them closely related to *C. kahawae*. These authors also reclassified an isolate from cranberry (CBS124.22) that was previously classified as *C. kahawae* subsp. *cigarro* [7], as *C. fructivorum*. Liu et al. [23] identified a new species, *C. jiangxiense*, among isolates from symptomatic and asymptomatic *Camellia sinensis* from China, phylogenetically closely related with CBD-causing isolates. Nevertheless, based on the pathogenicity tests and the pairwise homoplasy index test, these authors considered *C. jiangxiense* and CBD-causing isolates two independent species. Wang et al. [24] described, from diseased leaves of *Camellia sinensis* in China, a closely related species to *C. jiangxiense* and *C. kahawae* sensu lato, that was designated *C. wuxiense*. This later species can be distinguished from other species of the gloeosporioides complex using a concatenated ApMAT and *gs* gene tree [24]. More recently, Guarnaccia et al. [25] described *C. helleniense*, a species that is phylogenetically close to both subspecies of *C. kahawae* but clearly differentiated based on the sequence analysis of *cal*, *gapdh*, and *tub2* genes.

Host-shift speciation, a particular case of ecological speciation, is one of the main routes for the emergence of fungal pathogens [26,27]. An investigation on the origin, phylogeography, and evolution of the CBD-causing pathogen demonstrated that these fungi have undergone a recent speciation process via host-jump into coffee, exploiting the green coffee berries-ecological niche, departing from a background of a seemingly generalist group of fungi harmless to coffee berries [27]. While the divergence between these two groups is very recent (5600 years Before Present as an average estimate), genetic and biological evidences indicate the occurrence of an effective speciation event. These evidences include significant and elevated differentiation indexes across all studied loci and a complete segregation of polymorphic sites [27]. This process may have also shaped populations in other *Colletotrichum* spp. [28,29].

The combination of immigrant unviability, a strong intrinsic barrier to gene flow arising from the impossibility of reproduction between organisms that cannot infect the same host, with a predominantly asexual behavior (preventing the occurrence of recombination), would have been rather effective in keeping populations separated during the early stages of divergence, thus creating pleiotropic interactions between local adaptation and reproductive patterns to speed up the speciation process [27]. While the genealogical concordance criteria alone are insufficient to recognize these entities as two separate species [7,27], biological and population genetic data show that CBD-causing isolates and non-CBD causing isolates represent ecologically distinct and isolated groups that have separated quite recently, and can therefore be currently regarded as distinct species [13,27].

Additionally, Pires et al. [30] reinforced evidences of differentiation between these two taxa considering the genome size expansion of CBD-causing isolates in comparison to the closely related non-CBD causing isolates.

In this work, we combine previously published data on evolutionary and population genetics [27], genome size estimation [30] and population genomics [14] with new analyses of phenotypic characters and molecular data (including novel loci putatively associated to pathogenicity and to sexuality) to sustain that *C. kahawae* subsp. *kahawae* and *C. kahawae* subsp. *cigarro* should be recognized as distinct species, respectively *C. kahawae* and *C. cigarro comb. et stat. nov*.

## 2. Results

### 2.1. Pathogenicity Tests

All 32 CBD-causing isolates were pathogenic to green coffee berries, scoring above two in the 0–5 disease severity scale, and originating CBD symptoms. The remaining nine isolates were not capable of causing CBD, with a final disease severity score of zero (see Appendix A). For all isolates, conidia germination and appressoria formation rates were above 50%.

### 2.2. Induction of Perithecia

The isolates associated with CBD were not able to produce fertile perithecia either in homothallic or heterothallic crosses, even under long incubation periods (up to three months). However, several perithecia were observed for isolate Ang67, but these perithecia were not able to differentiate asci and ascospores, remaining immature in all the assays (Figure 1, panels A–D). Among the isolates not associated with CBD, isolates ICMP 18539 and ICMP 12953 produced fertile perithecia profusely on the toothpicks and on the growth medium from one week after inoculation onwards in homothallic crosses (Figure 1, panels F,G). For the remaining non-CBD causing isolates, perithecia were not observed.

### 2.3. Substrate Use

All non-CBD causing isolates, *C. camelliae* and *C. gloeosporioides* isolates metabolized at least citric acid or ammonium tartrate as a sole carbon source. The CBD-causing isolates were unable to use any of these substrates (see Appendix A).

### 2.4. Morphology

Colony diameter at different temperatures was recorded (see Appendix A). At 30 °C, all CBD-causing isolates grew significantly less (1 to 15 mm after 7 days of incubation) than the non-CBD causing isolates (16 to 37 mm for the closely related isolates and 63 to 61 mm for the *C. gloeosporioides* isolates). At the remaining temperatures, CBD-causing isolates tended to grow slower than non-CBD causing isolates, although less discriminately.

Conidia are cylindrical, straight with rounded ends. The average conidial length of CBD-causing isolates ranged between 11.7 and 18.1 µm, while that of non-CBD causing isolates ranged between 10.8 and 14.8 µm. The average conidial width ranged between 4.9 and 5.9 µm for the CBD-causing isolates and between 4.2 and 5.9 µm for non-CBD causing isolates. The length/width ratio ranged between 2.2 and 3.7 and between 2.1 and 3.4 for CBD-causing and non-CBD causing isolates respectively. Statistically significant differences were recorded between the set of CBD-causing isolates and the non-CBD causing isolates, with the former presenting longer (14.6 versus 13.2 µm) and wider (5.2 versus 5.1 µm) conidia, and with a larger length/width ratio (2.8 versus 2.7) (see Appendix A).

Hyphal appressoria are globose to fusiform, lobbed. The average appressorial length ranged between 9.9 to 11.8 µm for CBD-causing isolates and between 8.4 and 10.8 µm for non-CBD causing isolates, with a global average of 10.8 and 9.5 µm respectively, representing a statistically significant difference. The average appressorial width for CBD-causing isolates ranged between 5.9 and 7.7 µm (average 7.1 µm), while that for non-CBD causing isolates was of 5.2 to 7.6 µm (average 6.5 µm), again representing a statistically significant difference between both groups. The average length/width ratio ranged between 1.4 and 1.7 and between 1.2 and 2.0 for CBD-causing and non-CBD causing isolates respectively, with the appressoria of CBD-causing isolates more elongated (average 1.6) than those of non-CBD causing isolates (average 1.5), although with no significant differences.

### 2.5. Phylogenetic Analyses

The sequences for ApMAT, *tub2* and *gs* available in GenBank as *C. kahawae* sensu lato and sequences of isolates from related species, as *C. aotearoa*, *C. camelliae*, *C. clidemiae*, *C. fructivorum*, *C. jiangxiense*, *C. rhexiae*, *C. temperatum*, and *C. wuxiense* were retrieved and aligned with those obtained in the present study (see Appendix A). The ApMAT alignment has a length of 727 bases including alignment gaps and comprises 63 ingroup taxa. The *tub2* alignment has 599 bases including alignment gaps and 96 ingroup taxa, and the *gs* alignment has 877 bases including alignment gaps and comprises 40 ingroup taxa. The *C. gloeosporioides* isolate PR220 was used as outgroup. The tree topologies obtained by Bayesian consensus tree and Maximum Likelihood (ML) analyses were similar for ApMAT, *tub2*, and *gs* and therefore only Bayesian consensus trees are presented with bootstrap support values (BS) and posterior probability values (PP) near each node. Phylogenetic analyses of ApMAT identified monophyletic groups, each comprising isolates assigned to *C. aotearoa*, *C. camelliae*, *C. clidemiae*, *C. fructivorum*, *C. rhexiae*, *C. temperatum*, and *C. wuxiense* but is unable to separate *C. jiangxiense* from *C. kahawae* sensu lato (Figure 2, panel A). In the *gs* phylogeny, the CBD-causing isolates form a monophyletic group, and non-CBD causing isolates previously assigned to *C. kahawae* subsp. *cigarro* cluster in various groups (Figure 2, panel B). The *tub2* sequences were unable to resolve the species *C. fructivorum*, *C. jiangxiense*, *C. kahawae* subsp. *cigarro*, *C. kahawae* subsp. *kahawae*, and *C. rhexiae* (Figure 2, panel C).

The combined alignment of the nine loci (*apn25L*, ApMAT, *mat1-2-1*, *cas1*, *cellwall*, *siRNA*, *vosA*, *gs*, and *tub2*) comprises 8582 characters, including alignment gaps, from which 7656 characters were constant, 225 were parsimony-informative and 680 were variable but parsimony-uninformative. For each locus, the information of number of characters and parsimony informative sites, nucleotide substitution models, and score for the best tree of ML and Bayesian inference are listed in Table 1. The *C. gloeosporioides* isolate PR220 was used as outgroup. The topologies obtained by Bayesian consensus tree and ML analysis were identical for the single locus analysis, as well as in the nine gene-concatenated data.

The phylogenetic analysis of single loci of *apn25L*, *mat1-2-1*, *cas1*, *cellwall*, *gs*, *siRNA*, and *vosA* showed that the CBD-causing isolates cluster as a monophyletic group (with MPBS ≥70% and a Bayesian PP ≥ 0.95) and can be clearly separated from the non-CBD causing isolates, while ApMAT and *tub2* were unable to separate both groups of isolates. The *C. aotearoa* isolate was consistently placed out of the remaining. A similar situation occurred for the *C. camelliae* isolate except for *mat 1-2-1* and *gs* genes/regions. Isolates ICMP 18534 and CBS 237.49 formed a monophyletic group for all genes except for *gs*, *tub2*, and *vosA* (see Appendix A). The analysis of the nine genes/regions concatenated data set depicts the CBD-causing isolates as a monophyletic group, along with four groups comprising non-CBD causing isolates. The group phylogenetically closest to that of the CBD-causing isolates contains the isolates ICMP 18534 and CBS 237.49, and the holotype of *C. kahawae* subsp. *cigarro* (ICMP 18539) clusters in a basal position (Figure 3).

The Bayesian Poisson tree processes method (BPTP) model estimated seven species in our dataset (see Appendix A). The CBD-causing isolates are estimated to form a species with a PP of 0.83, while the non-CBD causing isolates are grouped in four hypothetical species, with PP ranging from 0.72 and 0.91.

### 2.6. In Silico Analyses of Proteins

The analysis of the predicted proteins obtained for cas1 showed that the seven CBD-causing isolates studied have two amino acid differences at positions 176 (D or N) and 178 (I or V), respectively, when compared with the proteins predicted for the six non-CBD causing isolates (Table 2). The CAS1 protein is an ortholog of an appressorium specific protein, with a DUF3129 domain (protein of unknown function), belonging to family Egh16-like virulence factor (IPR021476). This protein is predicted as extracellular (prediction confirmed by SignalP, TargetP, TMHMM, and Phobius), with a signal peptide and a cleavage site between position 22 and 23. Predictions of phosphorylation and glycosylation showed no differences between proteins from non-CBD or CBD-causing isolates as well as in the secondary structure.

Concerning the protein coded by the cellwall gene, the differences of the amino acid composition at position 42 (L or Q), 79 (K or Q), and 243 (P or L), and a deletion between positions 181 to 183 and from positions 204 to 223 can clearly distinguish CBD-causing isolates from non-CBD causing isolates (Table 2). The in silico analysis of structure/function of this protein showed the presence of the PFAM domain PF12296 (Hydrophobic surface binding protein A HsbA) and a signal peptide with a cleavage site between position 17–18 suggesting its extracellular localization (prediction confirmed by SignalP, TargetP, TMHMM, and Phobius). The kinase localization positions were predicted by NETPHOS (see Appendix A). It was observed that different phosphorylation sites were predicted in non-CBD causing isolates between positions 185–223 in the region that is absent from the CBD-causing isolates. The glycosylation predictor YinOYong suggested four glycosylation sites in non-CBD causing isolates located in positions 185, 210, 213 and 223 (see Appendix A). These sites are absent in CBD-causing isolates. The structural analysis of those proteins showed several differences, namely the percentage of disorder regions, the solvent accessibility and the secondary structure (Figure 4 and see Appendix A). In the VosA protein (viability of spores A), three amino acid differences at positions 49 (M or K), 128 (R or G), and 203 (P or A) were observed between the predicted proteins of CBD-causing isolates and non-CBD causing isolates. Predictions of phosphorylation and glycosylation showed no differences between proteins of the two groups of isolates as well as in the secondary structure.

For the siRNA protein, a putative argonaute siRNA chaperone complex subunit, ten amino acid differences were observed when comparing the 14 isolates listed in Table 2. However, when these isolates are grouped in CBD and non-CBD causing isolates, only one amino acid difference at position 194 (G to A) can distinguish both groups.

### 2.7. Genome Size

Genome size estimates ranged between 76.8 and 88.6 Mbp for CBD-causing isolates, and between 71.7 and 75.6 Mbp for non-CBD causing isolates. The genome size estimate for *Colletotrichum* sp. isolate PR428 is similar to that estimated for non-CBD causing isolates (72.9 Mbp). The *C. camelliae* isolate revealed the largest estimated genome size (88.7 Mbp) while the two *C. gloeosporioides* isolates have the smallest estimated genomes (average 70.3 Mbp). Statistical analysis performed on the genome size estimate of the isolates under study shows a significant difference between the CBD-causing isolates and non-CBD causing isolates. The average genome size of the CBD-causing isolates is 82.6 Mbp, while the group comprising non-CBD causing isolates previously assigned to *C. kahawae* subsp. *cigarro* shows an average genome size of 73.5 Mbp (see Appendix A).

### 2.8. Taxonomy

In this work, we have performed an analysis involving pathological, morphological, cytogenomic, biochemical, and molecular (new or re-assessed) traits that in combination shows that CBD-causing isolates can be clearly distinguished from the phylogenetically close non-CBD causing isolates. Based on the pathogenicity test on green coffee berries, metabolism of carbon sources, phenotypic characters such as growth rate at 30 °C, phylogeny, genome size, and population and evolutionary data from Silva et al. [27] and Vieira et al. [14], *Colletotrichum kahawae* subsp. *kahawae* revealed to form a separate species within the *Colletotrichum* genus and should be re-elevated to the species rank, *C. kahawae*. Consequently, the closely related fungi previously assigned to *C. kahawae* subsp. *cigarro* should also be raised to the species level.

*Colletotrichum cigarro* (B.S. Weir and P.R. Johnston) A. Cabral and P. Talhinhas, comb. et stat. nov. MycoBank MB 830326.

Basionym: *Colletotrichum kahawae* subsp. *cigarro* B.S. Weir and P.R. Johnst. [as *‘ciggaro’*] Studies in Mycology 73:115-180. 2012.

Holotype: Australia, on *Olea europaea*, coll. V. Sergeeva UWS124, 1989, PDD 102232; ex-type culture ICMP 18539.

Weir et al. [7] provide a description.

Note: Weir et al. [7] used the term “ciggaro”, in an attempt to reflect the Portuguese word for “cigarette”. However, the correct term in Portuguese is “cigarro”, and this fact was already been corrected in Mycobank MB 626870.

The growth rate on PDA at 30 °C can be useful to discriminate both entities, since *C. cigarro* isolates (non-CBD causing) grow more after seven days cultivation on PDA (16 to 37 mm) than *C. kahawae* (1 to 15 mm). The genes *apn25L*, *cas1*, *cellwall*, *gs*, *mat1-2-1*, *siRNA*, and *vosA* clearly separate *C. kahawae* from *C. cigarro*. The isolates studied that were previously assigned to *C. kahawae* subsp. *cigarro* by Weir et al. [7] are grouped phylogenetically into four clusters. As a growing number of isolates has regularly been assigned to this group, in the future there may be a basis to further dissect this taxon as a putative cryptic species complex.

## 3. Discussion

The species *C. kahawae* was created to accommodate the CBD pathogen, found only in Africa, providing a taxonomic tool to plant pathologists to name and distinguish this pathogen from other *Colletotrichum* strains inhabiting coffee but incapable of causing this disease [9]. In mycological terms, however, this taxon has been shadowed by neighbouring fungi clustering in *C. gloeosporioides* sensu lato [8]. While resolving species limits and phylogenetic relationships within *C. gloeosporioides* sensu lato, Weir et al. [7] established numerous new species, but the differences between the CBD pathogen and closely related fungi were considered to be insufficient to recognize them as separate species. The CBD pathogen was placed along with non-CBD causing isolates into two subspecies within *C. kahawae*. This taxonomic framework led, in recent years, to the identification of “*C. kahawae*” (no subspecies given) in other hosts and regions [32,33,34,35], raising concern among plant pathologists dealing with coffee diseases, especially when the spread of *C. kahawae* out of Africa is so feared [13].

The taxonomy of *Colletotrichum* is no longer supported exclusively by morphology and host-specificity, since a polyphagous behaviour is common in most *Colletotrichum* species complexes [36]. The recent advances in molecular biology allowed new DNA-sequence based tools leading to more accurate phylogenetic relationships. The combination of molecular and pathological data has evidenced a very recent host jump-based speciation event leading to the Coffee Berry Disease pathogen, *C. kahawae* [14,27].

In this work, we provide a formal framework to differentiate *C. kahawae* from its close relatives with the establishment of a newly designated species, *C. cigarro*. For this we have performed analyses involving morphological, pathological, biochemical, cytogenomic, and molecular data with a comprehensive set of CBD-causing isolates and phylogenetically close non-CBD causing isolates.

Combining pathological and molecular data, Silva et al. [27] demonstrated a clear, albeit recent, speciation process leading to the differentiation of the CBD pathogen. In the present work, additional molecular data obtained for an enlarged set of relevant isolates further strengthen that divergence. This study revealed that, besides the genes *apn25L*, *mat1-2-1*, and *gs*, the genes *cas1*, *cellwall*, *siRNA*, and *vosA* are also able to discriminate between CBD and non-CBD causing isolates. The phylogenetic analysis performed demonstrate that CBD-causing isolates are grouped in a monophyletic clade while the non-CBD causing isolates (designated by Weir et al. [7] as *C. kahawae* subsp. *cigarro*) grouped in four hypothetical species estimated by the BPTP model, that are clearly different from CBD-causing isolates. Similar results were obtained by Vieira et al. [14] that identified 9160 SNPs, which completely differentiated CBD-causing isolates from non-CBD causing isolates, and also observed long tree branches within clades, comprising the non-CBD causing isolates, suggesting a highly divergent group.

Moreover, the analysis of predicted proteins for *cas1*, *vosA*, and *cellwall* genes revealed differences in the amino acid composition between the two groups of isolates. These differences may affect the function of proteins and may be at the origin of the different pathological behaviour of each group of isolates, namely the ability to infect green coffee berries.

The CAS1 protein belongs to a group of fungal proteins that are found in pathogenic fungi and may play important roles during the early infection stage and appressoria development [37]; VosA protein contains a velvet domain with a DNA binding motif that recognize nucleotide consensus in the promotors of key developmental regulatory genes controlling several processes including toxin production, cell wall formation and the development of resting or sexual fruiting bodies [38]. In *Aspergilus nidulans*, VosA is required for sporogenesis and trehalose biogenesis of conidia and ascospores [39].

More significant seems to be the differences in amino acid composition of the cellwall protein, where the structural analysis prediction showed differences in the percentage of disorder regions, solvent accessibility and secondary structure along with different number and localization of phosphorylation and glycosylation sites in CBD-causing and non-CBD causing isolates. Although such sites can be reversibly and dynamically modified by O-GlcNAc or phosphate groups during cell life [40], these differences could be functionally significant. The *cellwall* gene studied here is an orthologue of the gene ENH85888 from *Colletotrichum orbiculare* that belongs to the top 100 most highly expressed genes at one-day post inoculation [41], and the differences observed in the amino acid sequence between CBD and non-CBD causing isolates may affect their virulence and pathogenicity. It has been previously referred a higher diversity on secreted and nuclear localized proteins that could play an important role in adaptation to lineage-specific infection lifestyles [42].

Overall, the differences in protein structures could be related with the differences observed in the pathogenic behaviour of the CBD pathogen, since the major differences were reported in proteins that should be related with the early steps of the infection process and hypothetically with the ability to infect green coffee berries. The predicted differences in the secondary structure observed between proteins of CBD-causing and non-CBD causing organisms suggest that their function may be altered or even compromised. This may be related to the differences observed in the infection process. In fact, structural phylogenetics provide an important complement to sequence-based analyses [43].

Morphological differences between CBD and non-CBD causing isolates concerning conidia and appressoria are scarce, as previously noted by Weir et al. [7]. The growth rate on PDA at 30 °C, however, was useful to discriminate both entities. Furthermore, while none of the CBD-causing isolates was able to differentiate fertile perithecia, some of the non-CBD causing isolates produced fertile perithecia readily and profusely. Among the latter is the *C. cigarro* holotype, isolate ICMP 18539. Interestingly, one in 32 CBD-causing isolates were able to differentiate perithecia, although infertile. The difference in the capacity to differentiate fertile perithecia between CBD and closely related non-CBD causing isolates is newly reported here and could be related to the higher genetic diversity of the non-CBD causing isolates as compared to CBD-causing isolates and to the adoption of an asexual behaviour of the latter in their adaptation to green coffee berries [27]. Additionally, the capacity to use citrate/tartrate as a sole carbon source [9] also remains as a distinctive trait between CBD and non-CBD causing isolates.

The ca. 8 Mbp genome size expansion between CBD and non-CBD causing isolates reported by Pires et al. [30] was corroborated by the results obtained in this study. Genome size variation is known to occur among species and species complexes in *Colletotrichum* [30] and could in fact reflect the effect of evolution on fungal genomes [44,45].

Altogether, these cytogenetic, molecular, morphological, biochemical, and pathological differences between the CBD and non-CBD causing isolates provide additional evidence to assign these entities into separate species. In fact, molecular data led recently to the differentiation of *C. camelliae*, *C. fructivorum*, *C. helleniense*, *C. jiangxiense*, *C. rhexiae*, *C. temperatum*, and *C. wuxiense* [22,23,24,25] from *C. kahawae* sensu lato. The populations that remained in *C. cigarro*, including the holotype ICMP 18539, are here recognized as separated from the CBD-causing pathogens, although further studies should be conducted with a larger collection of isolates from different hosts and locations to further dissect this taxon as a putative cryptic species complex.

## 4. Materials and Methods

### 4.1. Fungal Material

This study included monosporic isolates of *Colletotrichum* spp. associated with coffee and other hosts (Table 3). CBD-causing isolates from CIFC (Centro de Investigação das Ferrugens do Cafeeiro, Instituto Superior de Agronomia, Universidade de Lisboa, Lisboa, Portugal) collection were obtained from green *Coffea arabica* berries exhibiting CBD symptoms comprising 10 African countries representing the three genetic groups that were previously identified [27]: Angola, Cameroon, and East Africa (including Burundi, Ethiopia, Kenya, Malawi, Rwanda, Tanzania, Uganda, and Zimbabwe). Non-CBD causing isolates were selected in order to represent taxa closely related to *C. kahawae*, including isolates described as *C. kahawae* subsp. *cigarro* by Weir et al. [7] and as *Colletotrichum* sp. UG1 by Silva et al. [27], and one unassigned *Colletotrichum* sp. isolate (PR428), along with two *C. gloeosporioides* sensu stricto isolates (PR220 and PT808) [46].

### 4.2. Pathogenicity Tests

The isolates employed in this study were reassessed for their pathogenicity to green coffee berries. Detached expanding green berries of *Coffea arabica* variety Caturra (CIFC 19/1) were inoculated as previously described [47], incubated under saturating humidity at 22 °C and symptoms were evaluated every day up to 14 days after inoculation. Ten green berries were used for each isolate. Symptoms were scored based on a 0 to 5 scale [30], where 0—no symptoms, 1—discrete lesions (necroses) less than 2 mm in length, 2—lesions occupying less than 25% of berry surface, 3—lesions occupying half of the berry surface, 4—lesion occupying the entire berry surface, and 5—lesion with abundant sporulation. Isolates responsible for final average disease severity scores of 2 or higher were considered as pathogenic, while the remaining were not pathogenic.

Conidia germination and appressoria formation rates were obtained in vitro (on glass slides) using the spore suspension employed in the inoculation assay. The slides were placed in a humidity box at 22 °C for 22 h. After this period, observations were made using a light microscope (Leitz Dialux 20, Stuttgart, Germany).

### 4.3. Induction of Perithecia

The ability of each isolate listed in Table 3 to form perithecia was screened. Two 3 mm plugs, obtained from the actively growing margin of monosporic 10-day-old PDA cultures, were placed on opposite sides of 60 mm diameter petri dishes containing 6 mL of minimal salt medium (1.6% agar). On the medium surface three autoclaved birch toothpicks were placed in a “N” configuration to provide a substrate for the sexual structures [48]. The plates were incubated at 22 °C under a 12 h/day photoperiod and the formation of perithecia was checked up to 12 weeks after inoculation. Each experiment comprised three plates per isolate. The experiment was conducted twice.

Additionally, heterothallic crosses were conducted between all CBD-causing isolates as previously described with three repetitions per cross. The experiment was conducted twice.

### 4.4. Phenotypic Characters

Conidia, appressoria, perithecia, asci, and ascospores were measured and described according to Weir et al. [7] from monosporic cultures grown on Synthetic Nutrient-poor Agar medium [49] (SNA) at 20 °C under white fluorescent light with a 12 h/day photoperiod. The observations were done using a Leica DM 2500 microscope (Stuttgart, Germany) with differential interference contrast illumination and the images were captured using a Leica DFC295 digital camera using the software Leica Application Suite (LAS) version 3.3.0. For each informative structure, 30 measurements were obtained. Measurements are presented as (minimum–) first quartile–medium–third quartile (–maximum).

Cardinal temperatures for fungal growth were assessed by inoculating 90 mm petri dishes containing PDA with a 3 mm diameter plug cut from the edge of an actively growing colony. Growth was determined after 7 days in two orthogonal directions. Experiments were conducted at 5 to 35 °C with 5 °C intervals, with three replicate plates per strain at each temperature.

Quantitative morphological data were compared using the Tukey Honest Significant Difference mean comparison test at 95% confidence (STATISTICA 8.0, StatSoft Inc., Tulsa, OK, USA).

### 4.5. Substrate Use

The capacity to utilize citric acid or ammonium tartrate as sole carbon sources was evaluated [9] in 90 mm diameter plates, using two replicates per each isolate. The experiment was repeated twice. Plates were incubated at 25 °C and medium color registered after 7 days as purple (substrate used) or yellow (substrate not used).

### 4.6. DNA Extraction, PCR Amplification, and Sequencing

For each of the selected isolates, total genomic DNA was extracted from mycelia grown in PDA plates [50].

Gene sequences were obtained in this study or retrieved from GenBank from eight nuclear gene regions and one intergenic spacer: part of *Apn2* gene (*apn25L*), an intergenic spacer between the 3′ end of the *Apn2* gene and the mating type gene *mat1-2-1* (ApMAT), the mating type gene *mat1-2-1* (*mat1-2-1*), β-tubulin 2 (*tub2*) and glutamine synthetase (*gs*). Additionally, two genes were chosen from the *Colletotrichum orbiculare* transcriptome belonging to the top 100 most highly expressed genes at one day post inoculation [41], respectively: ENH76341, an appressorium specific protein (*cas1*); and ENH85888, a cell wall protein (*cellwall*). Additionally, two genes putatively involved in the sexual development [39,51] were used, a putative argonaute siRNA chaperone complex subunit (*siRNA*) and spore development regulator containing a velvet domain (*vosA*). Orthologous of each of the four genes were identified and sequenced for the isolates under study.

PCR amplifications were performed as previously described [20]. Primers used in this work are listed (see Appendix A). Sequencing in both directions was performed by StabVida (Portugal) and the assembly was done in the SeqMan module of DNASTAR software (Madison, WI, USA). All novel sequences were lodged in GenBank accession numbers MH346035 to MH346120, and alignments and phylogenetic trees in TREEBASE under reference TB2:S24331.

### 4.7. Phylogenetic Analyses

Sequences of the ApMAT, *gs*, and *tub2* previously available either from *Colletotrichum kahawae* sensu lato or from neighboring taxa, including *C. camelliae* Massee, *C. clidemiae* Weir and Johnst, *C. fructivorum* Doyle, Oudem, and Rehner, *C. helleniense* Guarnaccia and Crous, *C. jiangxiense* Liu and Cai, *C. rhexiae* Ellis and Everh., *C. temperatum* Doyle, Oudem, and Rehner and *C. wuxiense* Wang, Wang, and Yang [7,22,23,24,25] were download from Genbank (see Appendix A).

All the sequences obtained were used to performed two different analysis: one comparing all genes/regions sequenced for the isolates used in this study, and the other comparing these isolates with others, not physically handled in this study, using sequences retrieved from the GenBank for a set of genes limited by availability (namely *gs*, *tub2*, and ApMAT).

The DNA sequences were aligned using MAFFT version 7 [52] and the alignments edited manually, if necessary, in MEGA7 [53]. The alignments for each locus were combined in a single file using the program SequenceMatrix 1.8 [54]. Pryor to Bayesian inference, the best nucleotide substitution models for each locus were calculated in JMODELTEST 2.1.10 [55], with the following likelihood settings: number of substitution schemes = 3 (24 models), base frequencies (+F), proportion of invariable sites (+I), and rate variation among sites (+G) (nCat = 4). The models were selected according to the Akaike information criterion. MrBayes 3.2.6 [56] was used to perform the Bayesian analyses of the combined seven-loci dataset and individual locus data. The Markov Chain Monte Carlo sampling was set to 10 million generations, with two independent runs with four chains, one cold chain and three heated chains with a temperature of 0.2. The trees samples of the two cold chains were compared every 1000 generations and stopped when the average standard deviation of split frequencies fall below 0.01. Burn-in was set at 25% after which the likelihood values were stationary, and the remaining trees were used to calculate posterior probabilities. Trees from different runs were then combined and summarized in a majority rule 50% consensus tree. Maximum likelihood (ML) was implemented in the CIPRES Science Gateway V 3.357 [57] using RAxML-HPC 8 on XSEDE (8.2.9) [58] using the GTRCAT model and 1000 rapid bootstrap inferences.

Species boundaries were estimated using Bayesian Poisson tree processes method [59] (BPTP), using the nine-loci concatenated tree obtained in RAxML as input data. The calculations were conducted on the BPTP webserver, with 500,000 MCMC generations, with a thinning every 100 generations and a burn-in of 10%, the outgroup was excluded from the analysis. The convergence was checked by looking to the likelihood trace plot. The probability of each node to represent a species node was calculated using the maximum-likelihood solution.

### 4.8. Bioinformatic Prediction Tools

Gene predictions were performed in the program AUGUSTUS [60] training with the genome of *Fusarium graminearum* in order to compare the protein sequence from CBD-causing isolates and non-CBD causing isolates obtained for genes *cas1*, *cellwall*, *siRNA*, and *vosA*. The proteins obtained were aligned in MEGA7 [53], using MUSCLE and the amino acid differences were registered. The protein sequence analysis was done through a combination of INTERPRO [61], UNIPROT [62], HMMER [63], and PROSITE [64]. Secreted proteins were predicted using a battery of tools: SignalP (v4.1) [65], TargetP [66], and Phobius [67]. TMHMM [68] was used to exclude sequences with transmembrane domains. Predictions of serine, threonine, or tyrosine phosphorylation sites including generic and kinase specific were done by NETPHOS 3.1 [69]. Predictions for O-ß-GlcNAc attachment sites were performed in by YinOYang [70] server. GPS-LIPID [71] was used as a predictor for protein lipid modification sites. To predict and analyze protein structure, function, and mutations, the Protein Model Portal [72] and the PHYRE2 [73] servers were used.

### 4.9. Genome Size

The genome size was estimated for isolates listed in Appendix A. A Flow Cytometry protocol using propidium iodide-stained nuclei [30] was followed using the ascomycete *Cenococcum geophilum* Fr. (isolate 844.1, 1C = 0.208 pg/203 Mbp [74,75]) as the internal standard.

## 5. Conclusions

In this work, *C. kahawae* subsp. *cigarro* is renamed *C. cigarro*, whereas *C. kahawae* subsp. *kahawae* is re-elevated to the species status, in which only the isolates causing the Coffee Berry Disease are accommodated. Over the last few decades, taxonomic criteria in *Colletotrichum* have moved from morphology to molecular phylogeny. In this work we have combined phylogenetic and pathological data, among others, to demonstrate that high divergence is found at many levels supporting the recognition of distinct species, even though a remarkable genetic similarity shaped by a recent speciation event is still evident. Adding to the phylogenetic and biological relevance of these findings, the differentiation between *C. kahawae* and *C. cigarro* is of great importance for plant pathologists, plant breeders, and quarantine authorities to whom an accurate nomenclature is crucial to better distinguish CBD from non-CBD pathogens [4,5,6,7,8,9,10,11,12,13]. Since the causative agent of CBD, *C. kahawae*, is currently circumscribed to Africa, where annually determines heavy production losses, its spread around the world is a threat to the coffee production, leading for example the Australian and Chinese authorities to consider it as a quarantine pathogen. Resolving taxonomic ambiguities can be of relevance both to biology and agronomy, helping to improve food security [76], and distinguishing between the CBD pathogen, *C. kahawae*, and the cosmopolitan and polyphagous *C. cigarro* is such an example.

## Figures and Tables

**Figure 1 plants-09-00502-f001:**
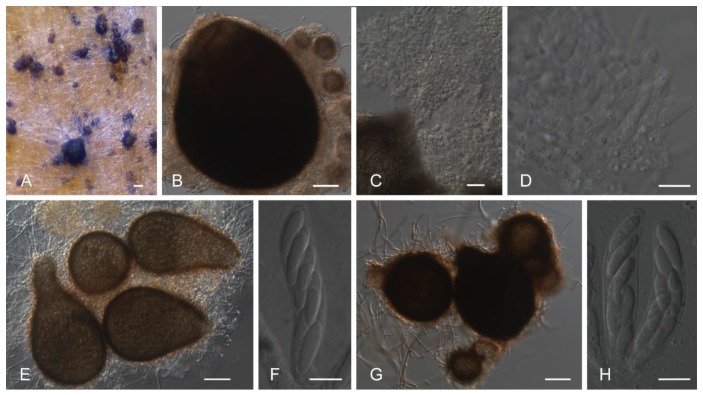
Differentiation of perithecia in *Colletotrichum* spp. *Colletotrichum kahawae* isolate Ang 67: (**A**,**B**), Perithecia observed after five weeks of incubation; (**C**,**D**), Perithecia oozing an undifferentiated mass. *Colletotrichum cigarro* isolates ICMP 18539 (**E**,**F**) and ICMP 12953 (**G**,**H**): E, G Perithecia; F, H Asci and ascospores. Scale bars A = 100 μm; B, E, G = 50 μm; C = 20 μm; D, F, H = 10 μm.

**Figure 2 plants-09-00502-f002:**
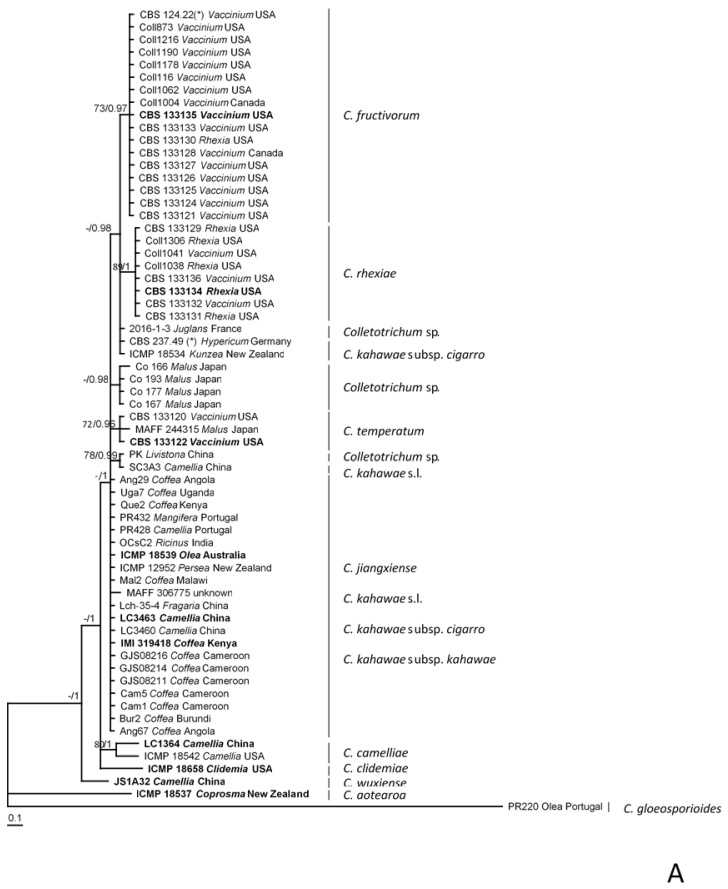
Fifty percent majority rule consensus tree from a Bayesian analysis based on the alignment the intergenic spacer between the 3′ end of the Apn2 gene and the mating type locus MAT1-2-1 (ApMAT) (**A**), of partial glutamine synthetase gene (*gs*) (**B**) and the β-tubulin gene (*tub2*) (**C**) enabling the comparison of the isolates under study with those with similar sequences publicly available. The RAxML bootstrap support (≥70) values (BS) and Bayesian posterior probability (PP; ≥0.95) are displayed at the nodes (BS/PP). The tree was rooted to *Colletotrichum gloeosporioides* (PR220). The scale bar indicates the expected changes per site. Ex-type cultures are emphasized in bold. (*) ex-type or authentic culture of synonymized taxon.

**Figure 3 plants-09-00502-f003:**
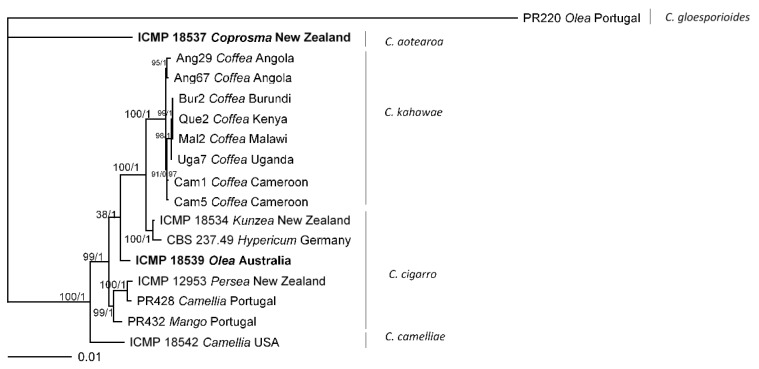
Fifty percent majority rule consensus tree from a Bayesian analysis based on a nine-loci combined dataset (*apn25L*, ApMAT, *mat1-2-1*, *cas1*, *cellwall*, *siRNA*, *vosA*, *gs*, and *tub2*) for isolates under study. The RAxML bootstrap support values (BS) and Bayesian posterior probability (PP) are displayed at the nodes (BS/PP). The tree was rooted with *Colletotrichum gloeosporioides* (PR220). The scale bar indicates the expected substitutions per site. Ex-type cultures are emphasized in bold. (*) ex-type or authentic culture of synonymized taxon.

**Figure 4 plants-09-00502-f004:**
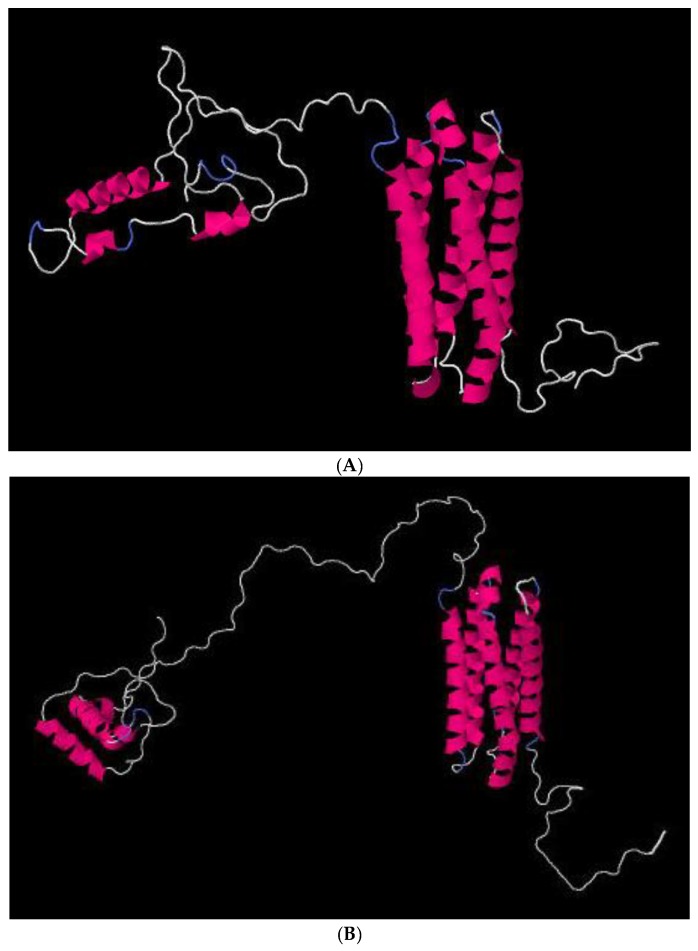
Molecular model (3D structure) of Cell wall protein of CBD-causing isolates (**A**) and non-CBD causing isolates (**B**) predicted by Phyre2. Jmol modelling software was used to visualize the predicted model [31].

**Table 1 plants-09-00502-t001:** Phylogenetic Information on the loci used in this study.

Loci ^1^	Taxa	Nucleotide Substitution Models	Chars	Constant	Parsimony-Informative	Parsimony-Uninformative	ML-ln L ^2^	BI-ln L ^3^
apMAT	64	K80	727	595	20	110	−1707.61	−1826.67
*gs*	47	HKY + G	877	751	39	68	−1941.59	−2021.35
*tub2*	96	HKY + G	599	541	19	31	−1180.52	−1289.96
*apMAT*	17	K80	725	614	12	98	−1528.87	−1588.01
*apn25L*	17	GTR	837	749	17	71	−1625.35	−1693.53
*cas1*	17	GTR + G	845	760	25	59	−1647.75	−1737.89
*cellwall*	17	GTR + I	1239	1112	50	73	−2481.99	−2521.34
*gs*	17	HKY + G	859	785	18	53	−1601.82	−1640.14
*mat1-2-1*	17	GTR + G	843	786	10	47	−1465.25	−1512.12
*siRNA*	17	HKY + G	1292	1112	52	125	−2839.28	−2880.50
*tub2*	17	HKY	597	548	8	33	−1067.93	−1100.61
*vosA*	17	HKY + G	1345	1190	33	121	−2723.89	−2760.43
combined	17		8582	7656	225	680	−17098.40	−17350.83

^1^ apMAT—an intergenic spacer between the 3′ end of the Apn2 gene and the mating type gene mat1-2-1; apn25L—part of Apn2 gene; cas1—appressorium specific protein; cellwall—cell wall protein; gs—glutamine synthetase; mat1-2-1—mating type gene; siRNA—a putative argonaute siRNA chaperone complex subunit; tub2—β-tubulin and vosA—developmental regulator; ^2^ ML-ln L—Likelihood score calculated in RAxML under GTRCAT model; ^3^ BI-ln L—Estimated marginal likelihoods from MrBayes.

**Table 2 plants-09-00502-t002:** Amino-acid differences observed in the predicted proteins of the *Colletotrichum* isolates for the genes *cas1*—appressorium specific protein; *cellwall*—cell wall protein; *siRNA*—a putative argonaute siRNA chaperone complex subunit and *vosA*—developmental regulator. Numbers refers to the amino acid position in the protein alignment. Isolates in bold-denote type strains. (*) = ex-type or authentic culture of synonymized taxon. Coffee Berry Disease (CBD), CBS-causing isolates, *Colletotrichum kahawae*; non-CBD, non-CBD causing isolates, *Colletotrichum cigarro*.

Identity	Isolate	Protein/Amino Acid Position in the Protein Alignment																																										
		cas1		siRNA										vosA					Cellwall																									
		176	178	71	85	101	146	194	227	236	281	306	322	49	94	128	203	309	42	79	181	182	183	208	209	210	211	212	213	214	215	216	217	218	219	220	221	222	223	224	225	226	227	243
CBD	Ang29	D	I	E	V	D	F	G	E	D	Y	S	D	M	V	R	P	L	L	K	-	-	-	-	-	-	-	-	-	-	-	-	-	-	-	-	-	-	-	-	-	-	-	P
	Ang67	D	I	E	V	D	F	G	E	D	Y	S	D	M	V	R	P	L	L	K	-	-	-	-	-	-	-	-	-	-	-	-	-	-	-	-	-	-	-	-	-	-	-	P
	Bur2	D	I	E	V	D	F	G	E	D	Y	S	D	M	V	R	P	L	L	K	-	-	-	-	-	-	-	-	-	-	-	-	-	-	-	-	-	-	-	-	-	-	-	P
	Cam1	D	I	E	V	D	F	G	E	D	Y	S	D	M	V	R	P	L	L	K	-	-	-	-	-	-	-	-	-	-	-	-	-	-	-	-	-	-	-	-	-	-	-	P
	Cam5	D	I	E	V	D	F	G	E	D	Y	S	D	M	V	R	P	L	L	K	-	-	-	-	-	-	-	-	-	-	-	-	-	-	-	-	-	-	-	-	-	-	-	P
	Mal2	D	I	E	V	D	F	G	E	D	Y	S	D	M	V	R	P	L	L	K	-	-	-	-	-	-	-	-	-	-	-	-	-	-	-	-	-	-	-	-	-	-	-	P
	Que2	D	I	E	V	D	F	G	E	D	Y	S	D	M	V	R	P	L	L	K	-	-	-	-	-	-	-	-	-	-	-	-	-	-	-	-	-	-	-	-	-	-	-	P
	Uga7	D	I	E	V	D	F	G	E	D	Y	S	D	M	V	R	P	L	L	K	-	-	-	-	-	-	-	-	-	-	-	-	-	-	-	-	-	-	-	-	-	-	-	P
non-CBD	ICMP 12953	N	V	D	V	E	F	A	D	E	K	S	D	K	V	G	A	H	Q	Q	A	T	P	P	A	T	P	K	T	P	A	A	A	P	A	A	P	K	T	P	A	A	A	L
	ICMP 18534	N	V	E	I	D	L	A	E	D	Y	S	D	K	V	G	A	L	Q	Q	A	T	P	P	A	T	P	K	T	P	A	A	A	P	A	A	P	K	T	P	A	A	A	L
	**ICMP 18539**	N	V	D	V	E	F	A	E	E	K	S	G	K	V	G	A	L	Q	Q	A	T	P	P	A	T	P	K	T	P	A	A	A	P	A	A	P	K	T	P	A	A	A	L
	CBS 237.49 (*)	N	V	E	I	D	L	A	E	D	Y	S	D	K	A	G	A	L	Q	Q	A	T	P	P	A	T	P	K	T	P	A	A	A	P	A	A	P	K	T	P	A	A	A	L
	PR432	N	V	D	V	E	F	A	E	E	K	P	G	K	V	G	A	L	Q	Q	A	T	P	P	A	T	P	K	T	P	A	A	A	P	A	A	P	K	T	P	A	A	A	L
	PR428	N	V	D	V	E	F	A	D	E	K	S	D	K	V	G	A	L	Q	Q	A	T	P	P	A	T	P	K	T	P	A	A	A	P	A	A	P	K	T	P	A	A	A	L

**Table 3 plants-09-00502-t003:** List of *Colletotrichum* spp. isolates analysed. Isolates in bold-denote type strains. (*) = ex-type or authentic culture of synonymized taxon.

Isolate	Species	Host	Country, Region
Ang6	*C. kahawae*	*Coffea arabica*	Angola, Chianga
Ang29	*C. kahawae*	*C. arabica*	Angola, Ganda
Ang30	*C. kahawae*	*C. arabica*	Angola, Ganda
Ang67	*C. kahawae*	*C. arabica*	Angola, Ganda
Ang81	*C. kahawae*	*C. arabica*	Angola, Huambo
Bur2	*C. kahawae*	*C. arabica*	Burundi
Cam1	*C. kahawae*	*C. arabica*	Cameroon, Babadjou
Cam2	*C. kahawae*	*C. arabica*	Cameroon, Santa
Cam5	*C. kahawae*	*C. arabica*	Cameroon, Baham
Cam8	*C. kahawae*	*C. arabica*	Cameroon, Kumbo
Eti3	*C. kahawae*	*C. arabica*	Ethiopia, Sidamo
Eti9	*C. kahawae*	*C. arabica*	Ethiopia, Sidamo
Eti20	*C. kahawae*	*C. arabica*	Ethiopia
Mal2	*C. kahawae*	*C. arabica*	Malawi
Que2	*C. kahawae*	*C. arabica*	Kenya
Que42	*C. kahawae*	*C. arabica*	Kenya
Que48	*C. kahawae*	*C. arabica*	Kenya, Taita Taveta
Que72	*C. kahawae*	*C. arabica*	Kenya, Ruiru
Que82	*C. kahawae*	*C. arabica*	Kenya, Kitale
Que84	*C. kahawae*	*C. arabica*	Kenya, Mgumguri
Rua1	*C. kahawae*	*C. arabica*	Rwanda, Gicumbo
Tan2	*C. kahawae*	*C. arabica*	Tanzania, Mbinga
Tan13	*C. kahawae*	*C. arabica*	Tanzania, Mbinga
Uga2	*C. kahawae*	*C. arabica*	Uganda, Kapchorwa
Uga3	*C. kahawae*	*C. arabica*	Uganda, Kapchorwa
Uga5	*C. kahawae*	*C. arabica*	Uganda, Kapchorwa
Uga6	*C. kahawae*	*C. arabica*	Uganda, Kapchorwa
Uga7	*C. kahawae*	*C. arabica*	Uganda, Kapchorwa
Uga9	*C. kahawae*	*C. arabica*	Uganda, Kapchorwa
Zim1	*C. kahawae*	*C. arabica*	Zimbabwe, Hiton
Zim12	*C. kahawae*	*C. arabica*	Zimbabwe
Zim14	*C. kahawae*	*C. arabica*	Zimbabwe
CBS 237.49 (*), ICMP 17922, C1275.8	*C. cigarro* (syn. *Glomerella cingulata* var. *migrans*)	*Hypericum perforatum*	Germany
ICMP 12953, C1206.3	*C. cigarro*	*Persea americana*	New Zealand
ICMP 18534, C1252.12	*C. cigarro*	*Kunzea ericoides*	New Zealand
**ICMP 18539, C1262.12**	*C. cigarro*	*Olea europaea*	Australia
ICMP 18542, C1291, CG02g	*C. camelliae*	*Camellia* sp.	USA
PR432	*C. cigarro*	*Mangifera indica*	Portugal, Lisbon
PR428	*Colletotrichum* sp.	*Camellia japonica*	Portugal, Lisbon
PR220	*C. gloeosporioides* s.s.	*O. europaea*	Portugal, Tondela
PR808	*C. gloeosporioides* s.s.	*O. europaea*	Portugal, Silves

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
