# Peer review of "Pathological, Morphological, Cytogenomic, Biochemical and Molecular Data Support the Distinction between Colletotrichum cigarro comb. et stat. nov. and Colletotrichum kahawae"

_plants, 2020, doi:10.3390/plants9040502_

Round 1
Reviewer 1 Report
This manuscript is rigorous, methodical and interesting. It is mostly suitable for publication in current form. However, as someone with a background more oriented to pathology and pharmacology, my interests in phylogeny slant more toward mechanistic interpretations than what the paper has provided. Consequently, my request would be to delineate possible hypotheses for why specific observed differences in the physiology of CBD versus non-CBD (i.e., CAS1, VosA, citrate/tartrate processing, etc.) actually translate into coffee bean pathology.
I think it is reasonable for the hypotheses to remain unresolved at this stage. After all, the mere act of questioning of why a specific difference is a functional differentiator can be useful for both honing practical applications as well as performing conceptual validation.
Beyond this, my only minor comments involve a some phrasing glitches in lines 37 and 38:
"... along the last decades." -- standard phrasing would be something like, "... in recent decades."
"Since its creation, and for over one century..." -- Presuming that "it" refers to "Colletotrichum", it is invalid to say that it was 'created', since this implies that an actual material synthesis. Instead of 'creation', it would be proper to say 'identification', 'naming', or 'initial description'. Also, the construct "and for over one century" is awkward. Somewhat better would be to say, "and through the next century".
Author Response
This manuscript is rigorous, methodical and interesting. It is mostly suitable for publication in current form. However, as someone with a background more oriented to pathology and pharmacology, my interests in phylogeny slant more toward mechanistic interpretations than what the paper has provided. Consequently, my request would be to delineate possible hypotheses for why specific observed differences in the physiology of CBD versus non-CBD (i.e., CAS1, VosA, citrate/tartrate processing, etc.) actually translate into coffee bean pathology.
I think it is reasonable for the hypotheses to remain unresolved at this stage. After all, the mere act of questioning of why a specific difference is a functional differentiator can be useful for both honing practical applications as well as performing conceptual validation.
R: Further discussion was added (lines 391-393.)
Beyond this, my only minor comments involve a some phrasing glitches in lines 37 and 38:
"... along the last decades." -- standard phrasing would be something like, "... in recent decades."
R: corrected
"Since its creation, and for over one century..." -- Presuming that "it" refers to "Colletotrichum", it is invalid to say that it was 'created', since this implies that an actual material synthesis. Instead of 'creation', it would be proper to say 'identification', 'naming', or 'initial description'. Also, the construct "and for over one century" is awkward. Somewhat better would be to say, "and through the next century".
R: corrected
Reviewer 2 Report
Vey good work. Many experiments have been performed in the current study. The concept, discussion and conclusion sections are fine. Introduction may need to be modified. It is too long. The introduction could be shortened. In discussion and results parts it is suggested to cite any similar publications if there are. I did not find enough citations in results and discussion parts. Many tables and figures have been used in the current study. It is suggested to omit or shortened them. Much more information has been shown in Figure 2. The quality of figure 2 may be improved. The language needs to be polished. Very recent articles are suggested to be used. This would increase the quality of the work. The following articles are recommended to be cited in the current manuscript. 1. Vahid Farzaneh, Jorge Gominho, Helena Pereira, Isabel S. Carvalho. Screening of the Antioxidant and Enzyme Inhibition Potentials of Portuguese Pimpinella anisum L. Seeds by GC-MS. Food Analytical Methods. https://doi.org/10.1007/s12161-018-1250-x. 2018. 2. Vahid Farzaneh, Isabel S. Carvalho. Modelling of Microwave Assisted Extraction (MAE) of Anthocyanins (TMA). Journal of Applied Research on Medicinal and Aromatic Plants, https://doi.org/10.1016/j.jarmap.2017.02.005. 2017. 3. Vahid Farzaneh, Isabel S. Carvalho. A review of the health benefit potentials of herbal plant infusions and their mechanism of actions. Published in Industrial crops and products. https://doi.org/10.1016/j.indcrop.2014.10.057. 2015. Look forward to reviewing the revised version of the current manuscript.
Author Response
Vey good work. Many experiments have been performed in the current study. The concept, discussion and conclusion sections are fine. Introduction may need to be modified. It is too long. The introduction could be shortened.
R: We have removed some less fundamental parts of the introduction.
In discussion and results parts it is suggested to cite any similar publications if there are. I did not find enough citations in results and discussion parts. Many tables and figures have been used in the current study. It is suggested to omit or shortened them. Much more information has been shown in Figure 2. The quality of figure 2 may be improved. The language needs to be polished. Very recent articles are suggested to be used. This would increase the quality of the work. The following articles are recommended to be cited in the current manuscript. 1. Vahid Farzaneh, Jorge Gominho, Helena Pereira, Isabel S. Carvalho. Screening of the Antioxidant and Enzyme Inhibition Potentials of Portuguese Pimpinella anisum L. Seeds by GC-MS. Food Analytical Methods. https://doi.org/10.1007/s12161-018-1250-x. 2018. 2. Vahid Farzaneh, Isabel S. Carvalho. Modelling of Microwave Assisted Extraction (MAE) of Anthocyanins (TMA). Journal of Applied Research on Medicinal and Aromatic Plants, https://doi.org/10.1016/j.jarmap.2017.02.005. 2017. 3. Vahid Farzaneh, Isabel S. Carvalho. A review of the health benefit potentials of herbal plant infusions and their mechanism of actions. Published in Industrial crops and products. https://doi.org/10.1016/j.indcrop.2014.10.057. 2015. Look forward to reviewing the revised version of the current manuscript.
R: The results and discussion sections were carefully revised for syntax errors and accuracy of scientific content. A better quality version of Figure 2 is now supplied in a separate file. We did not find opportunities to include the recommended citations.